# Chemically identifying single adatoms with single-bond sensitivity during oxidation reactions of borophene

Linfei Li [1], Jeremy F. Schultz [1], Sayantan Mahapatra [1], Zhongyi Lu[1], Xu Zhang [2] & Nan Jiang [1✉]

The chemical interrogation of individual atomic adsorbates on a surface significantly contributes to understanding the atomic-scale processes behind on-surface reactions. However, it remains highly challenging for current imaging or spectroscopic methods to achieve such a high chemical spatial resolution. Here we show that single oxygen adatoms on a boron monolayer (i.e., borophene) can be identified and mapped via ultrahigh vacuum tip-enhanced Raman spectroscopy (UHV-TERS) with ~4.8 Å spatial resolution and single bond (B–O) sensitivity. With this capability, we realize the atomically defined, chemically homogeneous, and thermally reversible oxidation of borophene via atomic oxygen in UHV. Furthermore, we reveal the propensity of borophene towards molecular oxygen activation at room temperature and phase-dependent chemical properties. In addition to offering atomic-level insights into the oxidation of borophene, this work demonstrates UHV-TERS as a powerful tool to probe the local chemistry of surface adsorbates in the atomic regime with widespread utilities in heterogeneous catalysis, on-surface molecular engineering, and low-dimensional materials.

[1] Department of Chemistry, University of Illinois Chicago, Chicago, IL 60607, USA. [2] Department of Physics and Astronomy, California State University, Northridge, Northridge, CA 91330, USA. ✉email: njiang@uic.edu

The chemical interrogation of single adatoms on a surface represents the most fundamental level of surface chemistry investigations[1]. For example, the oxidation of materials typically begins with the adsorption of dissociated oxygen atoms on the surface; therefore, the identification of single oxygen adatoms and their local chemistry is of essential importance to understanding the complex atomic and molecular processes that actuate oxidation reactions[2,3]. Advanced scanning probe microscopies (SPM), including scanning tunneling microscopy (STM) and atomic force microscopy and their derivatives, offer a potential route to this goal by topographic characterizations with atomic spatial resolution[4–6]. However, these techniques possess minimal chemical sensitivity, thereby providing very limited capabilities to identify the heterogeneity of surface adsorbates and particularly their chemical bonding with the substrate. Conversely, traditional optical spectroscopic methods, such as X-ray photoelectron, Raman, and infrared spectroscopy enable the determination of chemical bonds and oxidation states[7–9], but their ensemble averaging measurements lack spatial resolution and thus cannot probe site-specific surface properties and behaviors. Potentially, these compromises can be overcome by tip-enhanced Raman spectroscopy (TERS), a tandem technique of SPM with optical spectroscopy[10–14]. By confining laser light at the atomic-scale SPM junction and taking advantage of plasmon-enhanced Raman scattering, TERS can interrogate chemistry in the angstrom-scale regime with single-bond sensitivity, which allows chemical bonds to be resolved and mapped chemically at the spatial limit[15–23]. As a result, TERS is a versatile tool to explore chemical reactions on surfaces with atomic-level chemical spatial resolution.

Herein, we use STM-based TERS to investigate the oxidation of a recently synthesized two-dimensional (2D) metal–boron monolayer (i.e., borophene) on Ag(111) with atomic and molecular oxygen in ultrahigh vacuum (UHV) conditions. Borophene is chosen as a prototypical example due to its high polymorphism and in-plane anisotropy stemming from the multicenter bonding characteristic of boron[24,25], which holds promise for diverse and tunable adsorption behaviors and oxidative properties. However, there is still no consensus regarding the oxygen reactivity of borophene in experiments due to the lack of chemical interrogations at the atomistic scale[26,27]. It has potentially confounded ongoing efforts to realize borophene-related device applications in catalysis, energy, and nanoelectronics[28]. Therefore, understanding and controlling the oxidation of borophene at the atomic level is of fundamental scientific interest and extensive technological importance.

In this study, we demonstrated that the chemically uniform and thermally reversible oxidation of borophene can be achieved using atomic oxygen. Single oxygen adatoms on borophene were identified via UHV-TERS with ~4.8 Å chemical spatial resolution and single bond (B–O) sensitivity. In contrast, molecular oxygen exposure resulted in the spontaneous dissociation of $O_2$ on borophene at room temperature, which corroborates the high reactivity of borophene to oxygen. Phase-dependent oxidation behaviors were revealed by preferential adsorption of dissociated atomic oxygen on the $\nu_{1/6}$ rather than the $\nu_{1/5}$ phase, suggesting discrete chemical properties of borophene structures. Furthermore, the STM-induced decomposition of a boron oxide nanocluster was identified by TERS line scan in contrast to the stability of oxygen adatoms under the same induction conditions, which indicates the high stability of oxidized borophene. In addition to elucidating the oxygen affinity of borophene and realizing atomically defined borophene oxides, this work demonstrates the potential of UHV-TERS for investigating site-resolved structural and chemical properties of surface species at the atomic level.

## Results

**UHV oxidation of borophene with atomic oxygen.** UHV oxidation of borophene is first demonstrated using atomic oxygen. Oxygen atoms are produced by cracking $O_2$ on a heated (~1700 °C) iridium filament and then introduced to borophene surfaces at room temperature, as illustrated in Fig. 1a. Figure 1b–d shows the STM topography and atomic structures of pristine $\nu_{1/6}$ phase borophene[27,29]. Upon exposure to 6 Langmuir (L, 1 L = $1.0 \times 10^{-6}$ torr s) of atomic oxygen, borophene sheets were dotted with oxygen adatoms, which showed significantly bias-dependent morphology with a broad apparent size distribution of 5–9 Å (Supplementary Fig. 1). In particular, due to the convolution of the topography and electronic effects in STM imaging, oxygen adatoms are invisible at high sample biases (Fig. 1e) and imaged as depressions (Fig. 1f) or protrusions (Fig. 1g) at low biases. To avoid confusion, we hereafter only present images where oxygen adsorbates appear as protrusions. Large-scale and high-resolution STM images (Fig. 1h and inset) depict uniformly sized features and a random distribution for the adsorbed oxygen atoms on $\nu_{1/6}$ borophene. The homogeneous oxidation of borophene realized herein suggests an avenue to tune the electronic, mechanical, and phonon properties of borophene as a function of the oxygen surface concentration[30–32]. Note that a few big particles are observed particularly at borophene edges in Fig. 1e–h, which could be attributed to boron oxide clusters as minority species (see Supplementary Discussion 1).

STM imaging cannot determine the local chemistry of surface adsorbates, such as chemical bonding with a substrate, since it lacks chemical sensitivity. In contrast, Raman spectroscopy is a compelling approach for characterizing chemical bond properties due to its direct correlation with the vibrational behaviors of bonds. Having revealed their topography with STM, we next chemically interrogated individual oxygen adatoms with UHV-TERS. Figure 2a presents a TERS line scan showing the evolution of Raman profiles while the STM tip moves across the interface of an oxygen adatom on borophene. Remarkably, a prominent peak at 189 cm$^{-1}$ occurs when the tip is positioned on the borophene surface (e.g., pink plus and spectrum), which appears to be unshifted while the tip approaches the oxygen adatom. In contrast, two characteristic peaks located at 189 and 205 cm$^{-1}$ are observed upon localization of the tip on top of the oxygen adatom (e.g., red plus and spectrum), indicating the formation of new chemical bonds. In order to interpret the emerging Raman modes, we first determined the lowest energy configuration of oxygen adatoms on $\nu_{1/6}$ borophene with density functional theory (DFT) calculations. As shown in Fig. 2b–d, the most preferred adsorption sites of single oxygen atoms are bridge sites, in which an O atom links two adjacent boron atoms with identical covalent B–O bonds in conjunction with the cleavage of the B–B bond between the adjacent boron atoms (Fig. 2c, d). Based on the energetically favorable atomic structures, we carried out Raman simulations for pristine and oxygen-adsorbed $\nu_{1/6}$ borophene with the result shown in Fig. 2e–g. As previously demonstrated[33,34], the 189 cm$^{-1}$ peak acquired on bare borophene surfaces is readily assigned to the $B_{3g}^2$ Raman mode of $\nu_{1/6}$ borophene (186 cm$^{-1}$, Fig. 2e). The distinct Raman feature identified on the oxygen adatom is demonstrated to be oxygen-modified Raman modes based on a phonon simulation with the addition of an oxygen atom to $\nu_{1/6}$ borophene (Fig. 2f, g; see side views in Supplementary Fig. 4). The calculated frequencies for the two oxygen-modified Raman modes are 185 and 192 cm$^{-1}$, respectively. In particular, the former is almost identical to the frequency of the simulated $B_{3g}^2$ Raman mode of bare $\nu_{1/6}$ borophene (186 cm$^{-1}$). This simulation result is in excellent agreement with the experimental observation that the 189 cm$^{-1}$

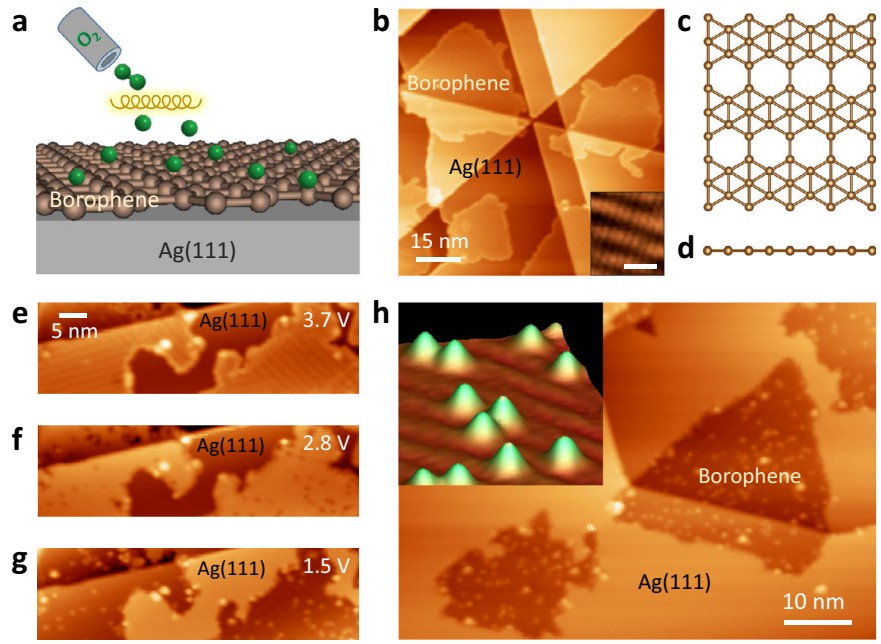

**Fig. 1 Oxidation of $v_{1/6}$ borophene with atomic oxygen. a** Schematic illustration of the experimental setup, with a heated iridium filament (yellow) used to produce atomic oxygen (green spheres) by cracking $O_2$ molecules. **b** STM image of as-prepared $v_{1/6}$ borophene sheets on Ag(111). Inset: atomic-resolution imaging of the $v_{1/6}$ borophene structure; scale bar: 2 nm. **c** Top and **d** side views of the atomic structure of $v_{1/6}$ borophene. **e**–**g** STM images of borophene sheets following exposure to 6 L of atomic oxygen, all of the same region with different scanning biases as indicated. Borophene islands are imaged as depressions in **g**. Uniformly shaped depressions (**f**) and protrusions (**g**) randomly distributed on borophene correspond to oxygen adatoms. **h** Large-scale oxidized $v_{1/6}$ borophene sheets. Inset: three-dimensional rendered STM image of oxygen adatoms on borophene. Scanning parameters: **b** 3.8 V, 200 pA; **e** 3.7 V, 100 pA; **f** 2.8 V, 100 pA; **g** 1.5 V, 100 pA; **h** 1.3 V, 200 pA.

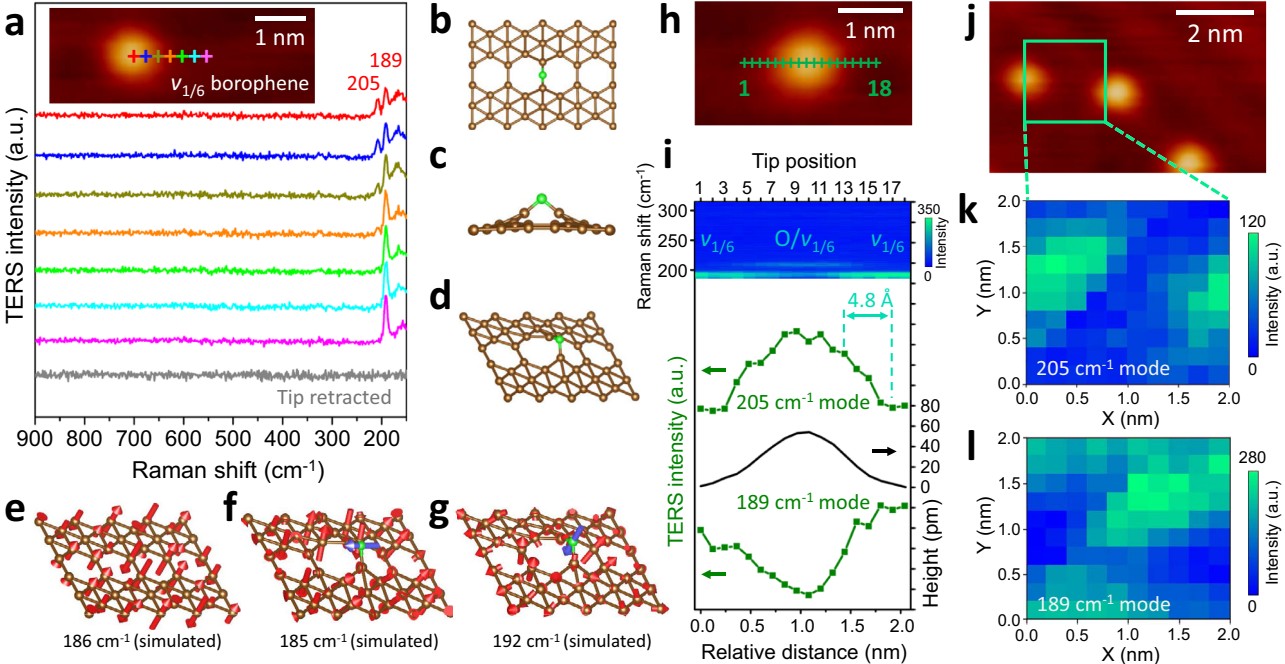

**Fig. 2 Angstrom-scale chemical identification of single oxygen adatoms on borophene via UHV-TERS. a** TERS line scan across the interface of an oxygen adatom on $v_{1/6}$ borophene along the trace marked with colored pluses in the inset STM image. **b** Top, **c** side, and **d** oblique views of the most energetically stable configuration of single oxygen adatoms (green spheres) on $v_{1/6}$ borophene. **e** Atomic displacements of the $B_{3g}^2$ Raman mode of pristine $v_{1/6}$ borophene. **f, g** Atomic displacements of Raman modes of O-adsorbed $v_{1/6}$ borophene. Green spheres and blue arrows represent oxygen atoms and their displacements, respectively. Waterfall plot of TERS line scan (**i**) across an oxygen adatom along the green trace shown in **h**. TERS intensity profiles (green) of the 205 and 189 cm$^{-1}$ modes and the corresponding height profile (black) of the oxygen atom are shown, indicating a chemical spatial resolution of ~4.8 Å. 2D TERS intensity mapping of the 205 cm$^{-1}$ (**k**) and 189 cm$^{-1}$ (**l**) modes over the region marked by a green square in **j**. TERS measurement parameters: **a** 0.1 V, 2 nA, 5 s acquisition time per point with a step length of 2.4 Å; **i** 0.1 V, 3 nA, 5 s per point with a step length of 1.2 Å; **k, l** 0.1 V, 3 nA, 3 s per pixel with a step length of 2.0 Å. All STM images were acquired at a sample bias of 1.5 V and a tunneling current of 300 pA. a.u., arbitrary units.

peak had an indiscernible shift when the tip moved from the borophene surface to the oxygen atom. Significantly, we found that the oxygen-modified Raman modes are not entirely localized on the oxygen adsorption site but are more delocalized and consist of vertical vibrations of single B–O bonds and adjacent B atoms. Consequently, site-resolved TERS measurements provide atomic-scale insights into the chemical bond between oxygen and borophene from its vibrational signatures. It is noteworthy that the simulations presented in this work were performed based on a freestanding borophene monolayer without taking into account interactions with the Ag substrate (e.g., charge transfer), which could be responsible for the small mismatch between theoretical values and experimental measurements (see Supplementary Discussion 2).

To see exactly how TERS spectra evolve across an oxygen adatom, we next performed 18 sequential TERS measurements along the line trace shown in Fig. 2h with a small step length of 1.2 Å. As seen in Fig. 2i, the waterfall plot of the TERS line scan exhibits a clear intensity evolution of the characteristic peaks at 189 and 205 $cm^{-1}$ across the oxygen atom. This demonstrates the capability to chemically identify a single oxygen adatom at the angstrom scale. Note that the evolution of the 189 $cm^{-1}$ mode reflects the Raman intensity change of both bare and O-adsorbed $\nu_{1/6}$ borophene. Significantly, the intensity profile of the 205 $cm^{-1}$ mode (green), which remarkably matches the trend found in the height profile of the oxygen atom (black), suggests a spatial resolution of ~4.8 Å (within a 10–90% contrast)[10]. Such a high spatial resolution allows adjacent atomic oxygen species to be resolved chemically. Figure 2k, l demonstrates this ability by mapping the 205 and 189 $cm^{-1}$ modes for two adjacent oxygen adatoms indicated in Fig. 2j, which are distinguished unambiguously in the TERS imaging. It is noteworthy that the shape of oxygen atoms in the TERS imaging appears more oval-shaped rather than a circle, which is most likely due to thermal drift during TERS collection (0.15–0.2 nm/min in X and Y directions under laser for our system)[23], a common phenomenon even at liquid-nitrogen temperatures (~78 K) particularly for an experiment which involves many data points to be measured (e.g., $10 \times 10$ pixels here).

**UHV oxidation of borophene with molecular oxygen.** In order to mimic the reactivity of borophene toward oxidation in air, we then interrogated the interaction of borophene with molecular oxygen at room temperature by exposing borophene sheets to $O_2$ in UHV (Fig. 3a). In contrast to atomic oxygen, exposure to comparable doses of $O_2$ molecules (e.g., 6 L) had almost no effect on borophene (Supplementary Fig. 5), which can be attributed to the relatively low reactivity of oxygen molecules compared to oxygen atomic radicals. However, upon exposure to substantial molecular oxygen (e.g., 3600 L) borophene sheets were subject to remarkable oxidation, as illustrated in Fig. 3b. In contrast to the topographically and chemically sharp edges of pristine borophene, high $O_2$ doses led to a significant morphological degradation of borophene edges by forming inhomogeneous oxide species, indicating the high sensitivity of borophene edges to molecular oxygen (see additional STM and TERS characterizations in Supplementary Fig. 6). In addition to oxidized edges, oxygen adsorbates of various sizes were observed in the interior of borophene sheets.

By virtue of the ultrahigh chemical sensitivity and atomic-scale spatial resolution of UHV-TERS, we can readily identify and differentiate these surface species by site-specific vibrational spectroscopy and surface chemical mapping. In particular, the smallest uniform protrusions, such as the one indicated by the green arrow (Fig. 3b), were identified to be oxygen adatoms by

verifying the vibrational signature at the frequencies of 205 and 189 $cm^{-1}$ (Fig. 3c, green) in contrast to the sole Raman peak of 189 $cm^{-1}$ (Fig. 3c, red) acquired on the adsorbate-free borophene surface (Fig. 3b, red arrow). The presence of atomic oxygen following dosing molecular oxygen demonstrates the spontaneous dissociation of $O_2$ molecules on borophene surfaces at room temperature[35]. Therefore, we reveal a propensity of borophene towards $O_2$ activation, which suggests promising applications in heterogeneous oxidative processes, such as surface catalysis and electrocatalysis[36]. In contrast to uniform oxygen adatoms, other adsorbed oxygen species appeared to be converted to surface oxides of high morphological and chemical heterogeneity. For example, the two clusters marked by cyan and blue arrows in Fig. 3b present complex and distinctive Raman profiles compared to that of oxygen adatoms (Supplementary Fig. 7), suggesting distinctly different chemical structures and adsorption configurations. These are consistent with the high structural diversity of borophene oxides predicted theoretically[37,38]. Overall, these results demonstrate the high reactivity of borophene to oxidation and the resulting significant surface degradation, which accounts for the decomposition of bare borophene into boron oxides (referred to as $BO_x$ hereinafter) in ambient conditions[26,39].

In this context, TERS is capable of identifying the heterogeneity of adsorbate species on oxidized borophene surfaces. A 2D TERS map of the intensities of the 205 $cm^{-1}$ mode over the white-box region is plotted in Fig. 3d that seemingly includes two kinds of adsorbates with different apparent heights. As a result, three oxygen adatoms as marked by the green arrows are explicitly identified as bright data pixels in the map, in contrast to the higher cluster indicated with the blue arrow that appears featureless at this Raman frequency (see typical point spectra of the cluster and oxygen adatoms in Supplementary Fig. 8).

Notably, the chemically distinct surface adsorbates that formed by exposure to molecular oxygen are in stark contrast to the uniform oxygen adatoms on atomic-oxygen-oxidized borophene. This remarkable difference could be attributed to varying adsorption configurations of $O_2$ molecules on borophene[35], which serve as heterogeneous nucleation sites for oxygen or oxide cluster formation. The situation is further complicated by multi-center boron–boron bonding configurations and anisotropic features of borophene. These result in the spatially inhomogeneous adsorption of a variety of complex oxygen species and oxide structures[32,37,38]. By contrast, atomic oxygen is structurally simple and has no variations in terms of adsorption orientations, which minimizes the likelihood of side reactions and byproducts and thus facilitates highly homogeneous chemical functionalization via uniform covalent attachment.

**Phase-dependent oxidation behaviors of borophene.** In addition to the aforementioned $\nu_{1/6}$ phase, $\nu_{1/5}$ is another prominent borophene polymorph with a distinct topography and atomic structure (Fig. 4a, b and Supplementary Fig. 9). In particular, $\nu_{1/5}$ domains feature prevalent self-assembled $\nu_{1/6}$ line defects[40], as shown in Fig. 4a and Supplementary Fig. 10. Similar to the oxidation of the $\nu_{1/6}$ phase, $\nu_{1/5}$ borophene surfaces were decorated with oxygen adatoms after exposure to atomic oxygen at room temperature (Fig. 4c). TERS was performed to chemically identify single oxygen adatoms on $\nu_{1/5}$ borophene. As seen in Fig. 4d, two TERS peaks at 180 and 202 $cm^{-1}$ are shown when the STM tip is located on top of the oxygen atom (green arrow and spectrum). In contrast, a sole peak at 180 $cm^{-1}$ is acquired on the adsorbate-free surface (orange arrow and spectrum), which is ascribed to the $B_g^2$ mode of pristine $\nu_{1/5}$ borophene (171 $cm^{-1}$ simulated) as displayed in Fig. 4e[33,34]. DFT simulations showed that the TERS peaks at 180 and 202 $cm^{-1}$ originate from two oxygen-modified

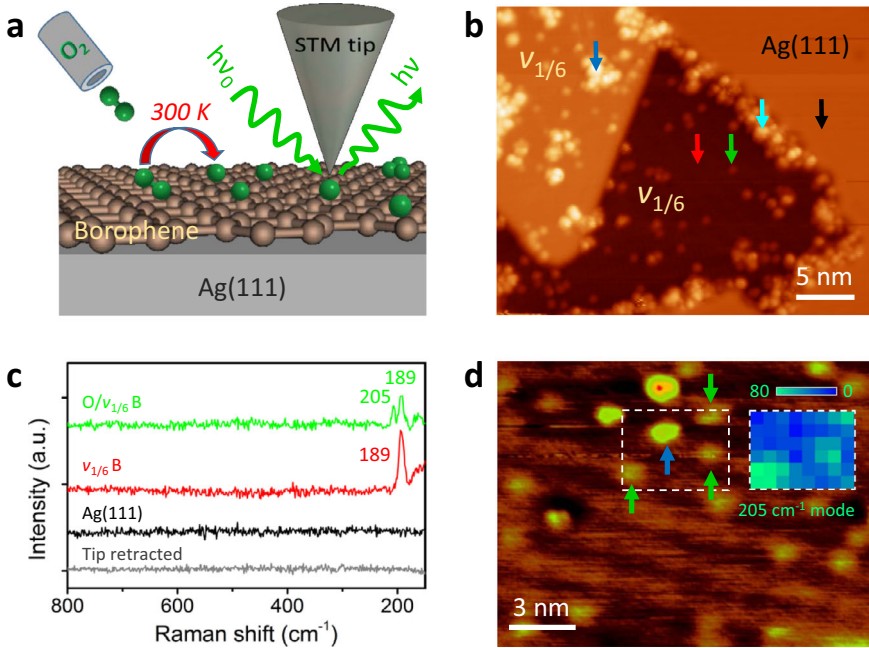

**Fig. 3 Oxidation of $v_{1/6}$ borophene with molecular oxygen. a** Schematic diagram of the experimental design. **b** STM image of a $v_{1/6}$ borophene island across a Ag(111) step edge after exposure to 3600 L of molecular oxygen. Arrows mark the tip positions for the TERS measurements presented in **c** and Supplementary Fig. 7. **c** TERS spectra acquired at the sites marked with green, red, and black arrows in **b**. **d** STM image of an oxidized borophene surface. Green arrows indicate three oxygen adatoms and a blue arrow marks a high cluster. 2D TERS map of the region marked by the white box is superimposed. TERS measurement parameters: **c** 0.1 V, 2 nA, 5 s; **d** 0.1 V, 3 nA, 2 s per pixel with a step length of 6.0 Å. Scanning conditions: **b** 0.5 V, 1.0 nA; **d** 1.3 V, 100 pA.

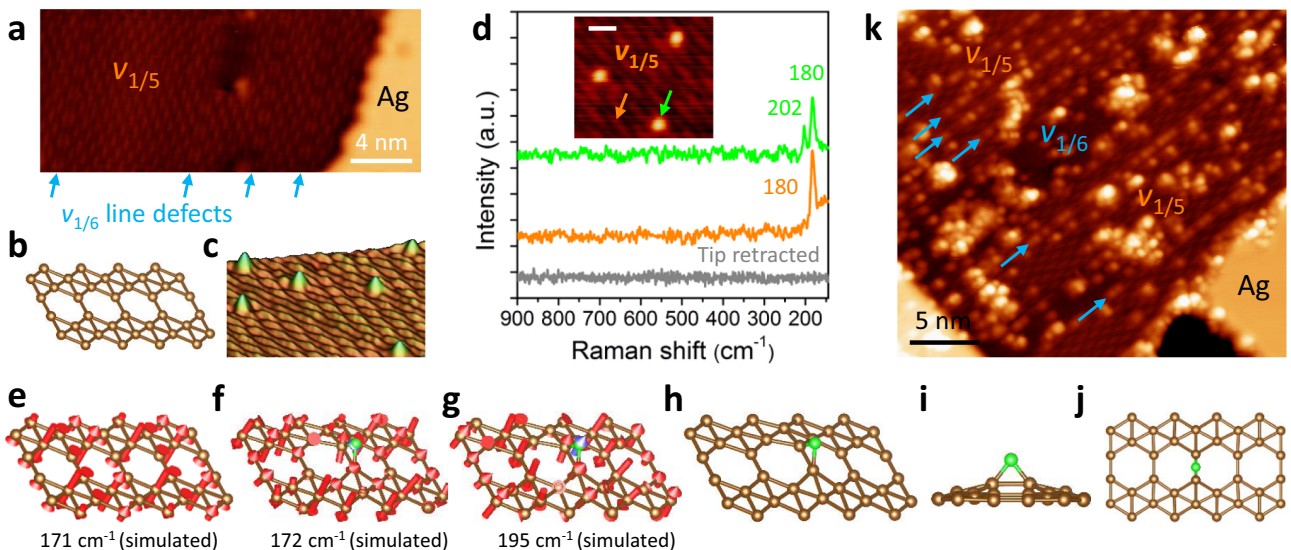

**Fig. 4 Oxidation of $v_{1/5}$ borophene with atomic and molecular oxygen. a** STM image of pristine $v_{1/5}$ borophene showing the presence of $v_{1/6}$ line defects as indicated by the blue arrows. **b** Oblique view of the atomic structure of $v_{1/5}$ borophene. **c** Three-dimensional rendered STM image of oxygen adatoms on $v_{1/5}$ borophene following exposure to 6 L of atomic oxygen. **d** TERS of atomic oxygen adsorbed on $v_{1/5}$ borophene. Inset: arrows mark the tip positions for TERS measurements; scale bar: 1 nm. **e** Atomic displacements of the $B_g^2$ Raman mode of pristine $v_{1/5}$ borophene. **f, g** Atomic displacements of Raman modes of O-adsorbed $v_{1/5}$ borophene. Green spheres and blue arrows represent oxygen atoms and their displacements, respectively. **h** Oblique, **i** side, and **j** top views of the most energetically stable configuration of an oxygen adatom on $v_{1/5}$ borophene. **k** STM image of $v_{1/5}$ borophene after exposure to 4800 L of molecular oxygen. A small $v_{1/6}$ domain and $v_{1/6}$ line defects indicated by blue arrows are present. TERS measurement parameters: 0.1 V, 3 nA, 5 s. Scanning conditions: **a** 1.0 V, 2.0 nA; **c** −0.2 V, 700 pA; **d** −0.2 V, 700 pA; **k** 0.8 V, 1.0 nA.

Raman modes with calculated phonon frequencies of 172 and 195 cm$^{-1}$, respectively, as illustrated in Fig. 4f, g. Particularly, the oxygen-modified 172 cm$^{-1}$ mode has a vibrational frequency very close to that of the oxygen-free $B_g^2$ mode (171 cm$^{-1}$), which

accounts for the experimental observation that the 180 cm$^{-1}$ peak appeared consistent while the tip moved from borophene to the oxygen adatom. The corresponding atomic structure models of an oxygen adatom on $v_{1/5}$ borophene are presented in Fig. 4h–j. A

distinct difference in oxygen adsorption configurations is that the two boron atoms bridged by the oxygen atom remain covalently bonded in $\nu_{1/5}$ rather than $\nu_{1/6}$ borophene (Fig. 4h, i), implying a weaker B–O bonding in oxidized $\nu_{1/5}$. As a result, the B–O–B ring composes a boron-based epoxide group, a configuration that has been identified in the epoxidation of graphene via the addition of atomic oxygen to graphene[41,42]. Consequently, TERS proves its ultrahigh chemical sensitivity by identifying single atoms adsorbed on different substrate structures by vibrational fingerprints. This could significantly contribute to comprehensive analyses of atomic and molecular adsorbates in chemically complex surfaces.

Notably, $\nu_{1/5}$ surfaces present distinctive morphology upon exposure to molecular oxygen (4800 L), as shown in Fig. 4k and Supplementary Fig. 11. In addition to the presence of aggregated amorphous $BO_x$ clusters, there is a preference for dissociated atomic oxygen adsorption along the $\nu_{1/6}$ line defects within $\nu_{1/5}$ domains, as indicated by the blue arrows. Closer inspection of the enlarged STM image of a mixed-phase borophene surface (Supplementary Fig. 12) further demonstrates that the adatoms preferentially adsorb on the $\nu_{1/6}$ phase rather than the $\nu_{1/5}$ phase, including $\nu_{1/6}$ domains and line defects. These observations are consistent with the TERS and DFT analyses that indicate stronger B–O interactions for the $\nu_{1/6}$ phase. Similar selective atomic adsorptions on borophene have been observed for hydrogen atoms[39], suggesting chemical specificity of different borophene phases. These results demonstrate discrete chemical properties of borophene structures, which can thus be modified and tailored by precisely tuning the polymorphism in borophene synthesis.

**Lateral mobility and thermal desorption of oxygen adatoms.** In addition to the adsorption energy, the mobility of atomic oxygen on the surface plays an important role in the oxidation process. Occasionally we observed the migration of individual oxygen adatoms under modest scanning conditions at ~78 K, as shown in Fig. 5a, b. Although this low-temperature observation is surprising given the strong B–O bonds found in oxidized borophene, it agrees with previous theoretical demonstrations that the lateral

displacement of oxygen atoms on borophene is feasible along a preferred diffusion path by overcoming a relatively low kinetic barrier[35]. Despite certain lateral mobility, the effective desorption of atomic oxygen from borophene only occurred at 360 °C, much higher than the desorption temperature of oxygen adatoms on UHV oxidized graphene (260 °C)[41], indicating the high stability of oxidized borophene due to strong B–O bonding. Importantly, atomic-oxygen-oxidized borophene was fully reduced at 360 °C resulting in a clean surface essentially identical to the pristine borophene surface, as shown in Fig. 5c, d. No discernible surface defects were observed following the desorption of chemisorbed oxygen, which can be rationalized by the high annealing temperature being within the typical temperature window for borophene growth (350–600 °C)[29,33]. The reduced borophene can be re-oxidized by exposure to atomic oxygen at room temperature, and homogeneous oxidized borophene surfaces similar to that shown in Fig. 5c can be recovered. Therefore, these observations demonstrate that the UHV oxidation of borophene using atomic oxygen is thermally reversible.

**Local interrogation of the stability of oxidized borophene.** In contrast to thermal stability measurements that rely on ensemble averaging, STM-TERS can investigate the stability of individual oxygen adsorbates at the atomic and molecular level[4,22], facilitating deeper insights into the local properties of oxidized borophene. Here we demonstrate this capability by studying the STM-tip induced decomposition of a single $BO_x$ nanocluster on oxidized borophene, as shown in Fig. 6. Figure 6a presents an oxidized $\nu_{1/5}$ borophene surface with three oxygen adatoms and a high $BO_x$ cluster. These surface adsorbates are stable under typical tunneling conditions below 3.8 V (Fig. 6b; oxygen adatoms are invisible due to the bias dependence of STM morphology). However, tip-induced decomposition happened to the $BO_x$ cluster when a 4.0 V scanning bias was applied. As shown in Fig. 6c and the blue height profile (Fig. 6i), the apparent size and height decreased abruptly after the upper half of the $BO_x$ cluster was imaged. Subsequently, a small and low cluster emerged, which was stable in the following image under the same scanning conditions (Fig. 6d). We can rule out the possibility of an imaging artifact recorded due to a sudden variation in tip conditions, because the features in the remainder of the images before and after the reaction appear substantially identical with no evidence of tip-induced morphologic changes. Additionally, this decomposition reaction was reproduced on another similar cluster under the same conditions (see Supplementary Discussion 3 and Supplementary Fig. 13). In contrast, following exposure to the elevated scanning conditions, the three oxygen adatoms remained intact and were reproduced with an imaging bias of 1.3 V (Fig. 6e), which suggests the high stability of oxygen adatoms on borophene. Note that similar elevated tunneling conditions have previously been found to lead to the desorption of oxygen adatoms from epitaxial graphene[41], indicating again the relatively strong B–O bonding compared to C–O bonding.

In order to further probe the stability of the boron oxide cluster, we attempted to break its internal bonding and extract atoms from it via STM manipulation, a protocol reported previously for metallic clusters[43,44]. Briefly, by approaching the tip very close to the $BO_x$ cluster with elevated tunneling current and scanning it, the cluster atoms could be extracted, as illustrated in Fig. 6g. Figure 6f presents the STM topography following exposure of the surface to an elevated tunneling current (1 nA). The $BO_x$ cluster is now absent likely due to attachment to the tip or displacement by the tip beyond the imaging scope. Instead, two low protrusions are present on the surface with an identical appearance to the three-preexisting oxygen adatoms. Notably, the latter remained steady after exposure to the elevated

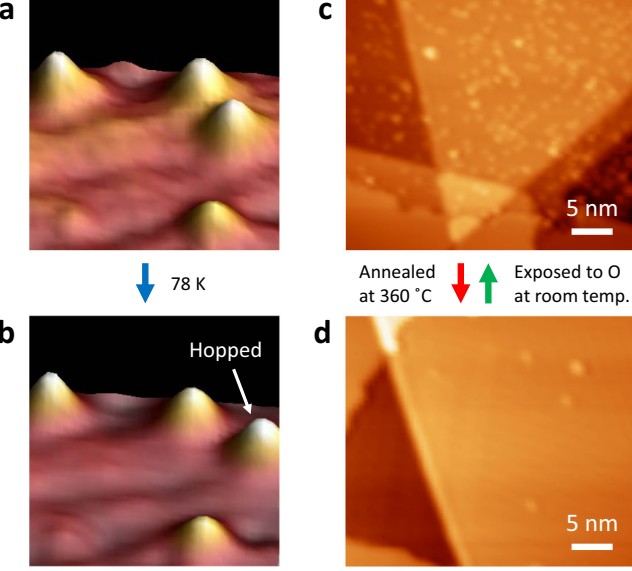

**Fig. 5 Thermal stability of oxidized borophene.** STM images **a** before and **b** after hopping of an oxygen adatom on borophene at low temperature (78 K). STM images of O-oxidized borophene **c** before and **d** after annealing at 360 °C, which were acquired from different surface areas. Scanning conditions: **a** 1.3 V, 100 pA; **b** 1.3 V, 300 pA; **c** 1.3 V, 100 pA; **d** 1.3 V, 150 pA.

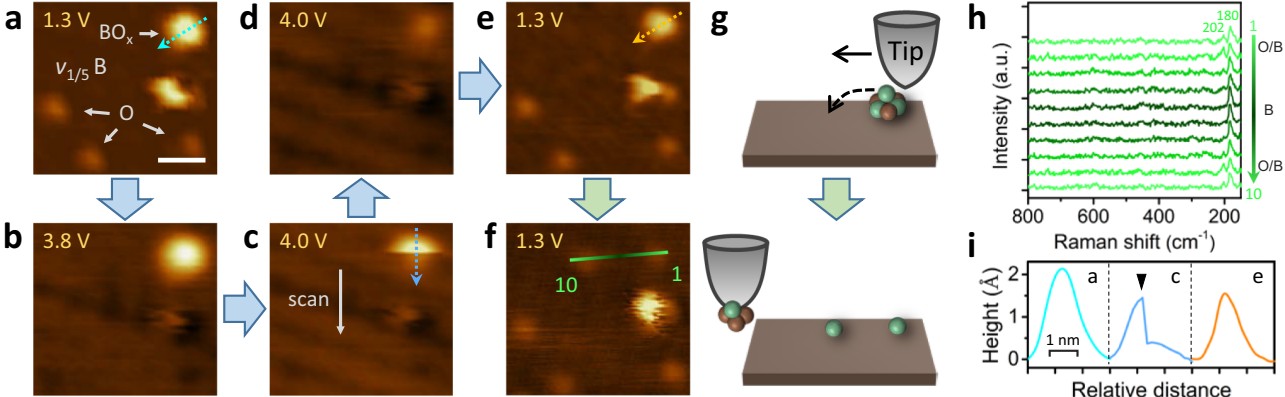

**Fig. 6 STM-TERS investigations of the stability of surface adsorbates on oxidized borophene. a–f** STM images acquired from the same oxidized borophene region with different sample biases as indicated. The absence of oxygen adatoms in **b–d** is due to the bias-dependence of STM topography rather than tip-induced desorption. White arrow in **c** indicates the STM scanning direction (top-down). Scale bar: 2 nm. **g** Schematic of tip manipulation performed in **e**, **f**. Brown and green spheres represent boron and oxygen atoms, respectively. **h** TERS line scan (10 points) along the green tip trace indicated in **f**. **i** Height profiles along the cyan (**a**), blue (**c**), and orange (**e**) dashed-line arrows. A black arrowhead marks the position of the decomposition reaction. TERS measurement parameters: 0.2 V, 2 nA, 5 s, step length ~4.5 Å. Tunneling current: **a–e** 150 pA; **f** 1 nA.

tunneling current. In order to identify the nature of these new features, we performed a TERS line scan across the protrusions. As plotted in Fig. 6h, the spectral modes at 202 and 180 cm$^{-1}$ are observed upon localization of the tip on top of the protrusions, thereby identifying these features as adsorbed oxygen atoms that are extracted from the previous cluster. Note that we can exclude the possibility that these fingerprinting TERS signals were from the potentially cluster-contaminated tip, because the same tip gave the sole Raman feature at 180 cm$^{-1}$ on the adsorbate-free borophene surface in between the two oxygen adatoms, which proves the spatial resolution of the TERS measurements and the cleanness of the tip in spectroscopy. Additional evidence is the featureless Raman spectrum acquired on the nearby Ag(111) surface under the same condition and the absence of Raman signals when the tip was withdrawn (Supplementary Fig. 14). These results further demonstrate the considerable stability of oxygen atoms in epoxide groups of oxidized borophene compared to those bonded within amorphous boron oxides under the same electronic energies.

## Discussion

In summary, we present atomic-level insights into the UHV-based oxidation of borophene via combined STM-TERS studies with angstrom-scale chemical spatial resolution and single-bond sensitivity. Controlled atomic oxygen exposure achieved the chemically homogeneous and reversible oxidation of borophene, which potentially imparts new electronic and chemical properties to borophene for nanoelectronic applications. Significantly, the resulting B–O–B groups could act as uniform nucleation sites for the growth of other materials on borophene using atomic layer deposition to realize borophene-based heterostructures and chemical functionalities[45]. Besides, UHV-TERS revealed the high chemical reactivity of borophene by spectroscopically identifying the dissociation of O$_2$ on borophene surfaces at room temperature. Distinct oxidation behaviors were observed on different borophene structures, opening an avenue to precisely tailor chemical properties and functionalities by modulating polymorphic variations in borophene synthesis. These chemical features of borophene promise a wide range of potential utilities in catalysis, sensors, and energy storage devices, but necessitate adequate protection from ambient oxidation and degradation[46]. We noticed that boron atoms with low coordination numbers

(e.g., 4 in $v_{1/6}$ or $v_{1/5}$ phase) and the bridge sites linking them show remarkable activity to atomic adsorption (e.g., H, O, F) via multi-centered-two-electron bonding or to the formation of covalent interlayer bonding (in multilayered borophene). This insight has been demonstrated by recent reports on the synthesis of hydrogenated borophene[39] and bilayer borophene[47] and their enhanced inertness to ambient oxidation due to the chemical passivation of active sites via B–H–B bonding or charge transfer and redistribution through covalent interlayer B–B bonds.

In addition, we demonstrate combined UHV-STM-TERS provides a comprehensive strategy to in situ interrogate adsorption behaviors with both topographic and chemical information describing interactions between adsorbates and the substrate. This work brings the spatial scale of optical spectroscopic investigations down to single adatoms and corresponding individual bonds with the substrate. The extension of this work to other atomic adsorbates beyond oxygen is expected to result in the investigation of highly localized surface chemistry in more complex nanoenvironments. For example, it is highly desirable to use UHV-TERS to determine the nature and transformation of chemical bonds at the atomic level in surface catalytic processes, which forms the critical foundation that controls the properties and performance of heterogeneous catalytic systems.

## Methods

**Preparation of oxidized borophene with atomic and molecular oxygen.**
All sample growth was carried out in a commercial UHV preparation chamber (~10$^{-10}$ torr) equipped with a standard molecular beam epitaxy experimental setup. Ag(111) foils on mica which were used as growth substrates were cleaned by several cycles of Ar$^+$ ion sputtering followed by annealing at 550 °C. Borophene was grown by electron beam evaporation (ACME Technology Co., Ltd) of a boron rod (99.9999% purity, ESPI Metals) onto the clean Ag(111) surface which was kept at 370 °C (for $v_{1/6}$ borophene), 450 °C ($v_{1/6} + v_{1/5}$), and 510 °C ($v_{1/5}$). UHV oxidation of borophene was realized by exposing as prepared borophene to molecular or atomic oxygen at room temperature. Oxygen gas was introduced by backfilling the UHV chamber via a leak valve. Atomic oxygen was produced with a hot iridium filament (~1700 °C) which was placed 50 mm from samples. Typical O$_2$ pressures during oxidation are 1 × 10$^{-5}$ torr (for molecular-oxygen oxidation) and 1 × 10$^{-7}$ torr (for atomic-oxygen oxidation), respectively. The total oxygen dose is described in Langmuirs (1 L = 1.0 × 10$^{-6}$ torr s).

**UHV-STM and TERS characterizations.** All STM and TERS measurements were performed at liquid nitrogen temperature (78 K) using electrochemically etched Ag tips in a UHV variable-temperature STM system (USM1400, UNISOKU Co., Ltd.) equipped with a home-built optical setup. A 561 nm solid-state CW laser (Lasos Laser GmbH) polarized parallel to the Ag tip was used as the excitation source with

a laser power of 6–8 mW used for TERS measurements. Significantly, the STM chamber is equipped with in vacuo lenses positioned in close proximity to the tip to allow for optimal laser spot focusing and collection efficiency[48,49]. Specific STM and TERS experimental parameters for each measurement have been indicated in the main text.

**Theoretical models**. To simulate the adsorption of oxygen on borophene, we employed a supercell constructed by $(10 \times 6)$ unit cells to model the $v_{1/6}$ and $v_{1/5}$ structures, which consist of 300 and 320 B atoms, respectively. The DFT calculations were carried out using the VASP package[50] with the projector augmented wave pseudopotentials[51] and the Perdew–Burke–Ernzerhof generalized gradient approximation[52]. An energy cutoff of 400 eV was used for the plane-wave basis set. Only the Γ-point was considered in the Brillouin zone due to the large size of the supercell, which had been tested to yield the converged results. All atoms were fully relaxed with the force convergence criterion being 0.01 eV/ Å. The phonon modes were obtained by solving the eigenvalue problems of a dynamical or Hessian matrix based on density functional perturbation theory calculations.

## Data availability

All data needed to support the findings of this study are present in the paper and/or the Supplementary Information. Additional data related to this study may be requested from the corresponding author.

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

## Acknowledgements

N.J. acknowledges support from the National Science Foundation (CHE-1944796). X.Z. acknowledges support from the National Science Foundation (DMR-1828019).

## Author contributions

L.L. and N.J. conceived the project. L.L. carried out experiments and data analyses with contributions from J.F.S., S.M., and Z.L. S.M. produced silver tips. X.Z. performed theoretical calculations and analyses. N.J. supervised the project. L.L. authored the manuscript in collaboration with all co-authors.

## Competing interests

The authors declare no competing interests.
