## [Peer Review File · Nature Communications]

Chemically identifying single adatoms with single-bond sensitivity during oxidation reactions of boropheneREVIEWER COMMENTS

Reviewer #1 (Remarks to the Author):

The work has beautifully demonstrated several capabilities of TERS that are deeply desired by surface scientists. 4.8Å spatial resolution of TERS images in an STM set up matches the state of the art reported before. Most demanding measurements reported in the article such as imaging bonds of single adatom on different phases of borophene will push the TERS technique forward and encourage utilisation of TERS in answering intriguing questions likes of oxygen reactivity with borophene (or other materials).

This body of work addressed a number of aspects beyond utilisation of TERS as a technique. Examples include, reactivity of atomic and molecular oxygen, differences in reactivities between different phases, evolution of TERS spectra across a bonded oxygen, hopping of an adatom captured using STM imaging, thermal desorption and stability of oxidised borophene.

Adequate detail of the work has been given in the main manuscript and in the supplementary information. The article reads well and figures are presented clearly.

The work is significant in pushing the sensitivity and spatial resolution of STM TERS as well as interrogating behaviour of adatoms using chemical signature such as bond vibrations using Raman spectroscopy.

Reviewer #2 (Remarks to the Author):

This manuscript by L. Li, et al. describes a novel and compelling demonstration of nanoscale chemistry measurements, utilizing STM-based TERS on oxidized borophene. In short, the experimental work is of excellent quality, I recommend it for publication enthusiastically, but I believe that there are two questions which this work is uniquely positioned to answer, only one of which would require any additional (very simple) experiments.

I will summarize my impressions of the work first. The authors have demonstrated synthesis of high quality samples of both primary borophene polymorphs on Ag(111). For each of these polymorphs, they perform STM-based cryogenic TERS. They identify Raman modes related to the pure and oxidized borophene for both phases. Oxidation is accomplished by both atomic O and molecular O₂ exposure. Differences in adsorption sites and oxidation behavior are highlighted between the v_{1/6} and v_{1/5} phases. Moreover, thermal reduction and tip-induced desorption are observed. The latter could easily be a platform for patterning!

The methods employed are appropriate. Technical details such as the use of the iridium filament are sound (to avoid potential WO_x deposition).

My questions (or perhaps, groupings of questions) are as follows:

(1) Can the authors more explicitly relate the local oxidation sites with the expected electronic structure of the borophene atoms? That is, given that the boron atoms in borophene are in a number of inequivalent coordinations, how does the observed oxidation relate to the expected stability of these various coordinations? Does this imply anything about the nature of bonding in borophene? And finally, does this suggest a route to optimize the stability of borophene in new polymorphs? (this might lead the field of borophene synthesis in interesting directions).

(2) Is there a dose limit for recovery after oxidation via atomic O or molecular O₂ exposure? Or restated differently, how much oxidation is fully destructive to the borophenes? The oxidation of borophene, and ostensibly the modification of the intrinsic properties, are a strong barrier to

further applications and straightforward device fabrication. It would be great to explore whether the borophene can be recovered from intense oxidation via a similar protocol detailed in Fig. 6.

Reviewer #3 (Remarks to the Author):

In this manuscript, the authors used ultrahigh vacuum tip-enhanced Raman spectroscopy (UHV-TERS) to chemically interrogate individual oxygen adatoms on a boron monolayer, which is helpful for the deep understanding of the atomic-scale reactions. It is nice to see that the authors successfully manage to manipulate single atom and induce the single atom reaction with the substrate, and to chemically identify the formation of the bond via vibrational fingerprint. The article can be published after addressing the following questions:

1. While discussing Figure 2, the authors described Figure a first, followed by Figure d and then Figure b, c (d comes before b and c, which is awkward). Thereafter, Figures e-g are described in the next paragraph. Such a logic is awkward. I highly suggest the authors to describe the phenomenon in Figure a first thoroughly, followed by simulation of the most stable structure and the possible vibrational modes. Then, the authors can try to correlate their result of simulation results.
2. In Figure 2i, from the description, it seems the authors assign both 205 and 189 cm^{-1} modes to the adsorbed single oxygen adatom. If this is the case, their trends of the TERS intensity profiles should not be inverse. I assume the authors would rather mean the 189 cm^{-1} mode comes from the mode of borophene under the influence of the neighboring O atom. If this is the case, the authors should clearly state this in the main text. Furthermore, the vibrational modes shown in Figure b-d can not be easily read. I would suggest the authors to increase the size or enlarge a portion of the model.
3. In the 2D TERS intensity mapping shown in Figure 2k and i, the authors mentioned that the oval-shaped rather a circle came from the thermal drift during TERS collection. To validate this assumption, the authors should provide the drift value of this TERS system, so that the result in the image distortion can be estimated.
4. In Figure 3d, the cluster shows no 205 cm^{-1} features in the 2D TERS mapping. It is better to also show the complete TERS spectra on the cluster site and the oxygen adatom site.
5. When comparing the activity of borophenes of different phases, why different amount of oxygen was introduced into the system (3600 L and 4800 L)?
6. Were Figures 5c and d obtained from the same position? I guess they are from different positions, as the substrate has undergone annealing. If this is the case, the authors should clearly state they are from different position, otherwise, the comparison of Figures 5a and b will mislead the understanding of c and d.
7. In figure 3b, why the two borophene with $\nu_{1/6}$ phase showed two different contrast?
8. It is not stated but can be observed that oxygen clustering tends to form near borophene edges, which need to be stated and explained.
9. The authors state that the simulations were performed based on a freestanding borophene monolayer without taking into account interactions with the Ag substrate, which led to the mismatch between theoretical values and experimental measurements. So, I am wondering are there any challenge in phonon simulations to include the effect of the Ag substrate.
10. In Figure 6c, when the bias is changed to 4.0 V, why the corresponding oxidized borophene height shows an immediate change at one line rather than changes from the beginning position?
11. The three isolated oxygen adatoms always exist in Figure 6a, e and f. It seems that the corresponding STM images show no features of these three adatoms when the bias is changed. Then, they will appear at the same position when the bias returned to 1.3 V. The authors should provide more explanations and the contrast bar of these STM images which is helpful for reading.
12. In Figure 6g, the author showed the scheme that there will be some clusters remaining on tip after the STM manipulation. We suggest the authors to check the contamination on the Ag(111) surface instead of clean borophene, which will be more convincing. The authors should also show the spectra of after tip retraction in all TERS results, which can convincingly demonstrate that the collected Raman signals are indeed from the sample instead of clusters on the tip.

In the point-by-point summary of the revisions below, the reviewers' original comments are italicized (in black), followed by our response (in blue) and corresponding changes to the manuscript (in green).

REVIEWER COMMENTS

Reviewer #1 (Remarks to the Author):

The work has beautifully demonstrated several capabilities of TERS that are deeply desired by surface scientists. 4.8Å spatial resolution of TERS images in an STM set up matches the state of the art reported before. Most demanding measurements reported in the article such as imaging bonds of single adatom on different phases of borophene will push the TERS technique forward and encourage utilisation of TERS in answering intriguing questions likes of oxygen reactivity with borophene (or other materials).

This body of work addressed a number of aspects beyond utilisation of TERS as a technique. Examples include, reactivity of atomic and molecular oxygen, differences in reactivities between different phases, evolution of TERS spectra across a bonded oxygen, hopping of an adatom captured using STM imaging, thermal desorption and stability of oxidised borophene.

Adequate detail of the work has been given in the main manuscript and in the supplementary information. The article reads well and figures are presented clearly.

The work is significant in pushing the sensitivity and spatial resolution of STM TERS as well as interrogating behaviour of adatoms using chemical signature such as bond vibrations using Raman spectroscopy.

Reply: We appreciate the reviewer's very positive comments and recommendation of publication.

Reviewer #2 (Remarks to the Author):

This manuscript by L. Li, et al. describes a novel and compelling demonstration of nanoscale chemistry measurements, utilizing STM-based TERS on oxidized borophene. In short, the experimental work is of excellent quality, I recommend it for publication enthusiastically, but I believe that there are two questions which this work is uniquely positioned to answer, only one of which would require any additional (very simple) experiments.

I will summarize my impressions of the work first. The authors have demonstrated synthesis of high quality samples of both primary borophene polymorphs on Ag(111). For each of these polymorphs, they perform STM-based cryogenic TERS. They identify Raman

modes related to the pure and oxidized borophene for both phases. Oxidation is accomplished by both atomic O and molecular O₂ exposure. Differences in adsorption sites and oxidation behavior are highlighted between the $v_{1/6}$ and $v_{1/5}$ phases. Moreover, thermal reduction and tip-induced desorption are observed. The latter could easily be a platform for patterning!

The methods employed are appropriate. Technical details such as the use of the iridium filament are sound (to avoid potential WO_x deposition).

Reply: We are grateful for the positive feedback from the reviewer, and we also appreciate the valuable comments and questions. We have responded to each of them below and revised the manuscript accordingly.

My questions (or perhaps, groupings of questions) are as follows:

(1) Can the authors more explicitly relate the local oxidation sites with the expected electronic structure of the borophene atoms? That is, given that the boron atoms in borophene are in a number of inequivalent coordinations, how does the observed oxidation relate to the expected stability of these various coordinations? Does this imply anything about the nature of bonding in borophene? And finally, does this suggest a route to optimize the stability of borophene in new polymorphs? (this might lead the field of borophene synthesis in interesting directions).

Reply: Boron has three valence electrons with an electronic configuration of [He]2s²2p¹, but has four valence orbitals. That is, there are not enough electrons to fill all the electronic orbitals in the chemical bonding of boron atoms, in contrast to the case of carbon systems (e.g. graphene) that can adopt normal two-centered-two-electron bonds and form a hexagonal (honeycomb) carbon lattice with sp² bonding. Consequently, boron-related crystals are prone to multiple-centered- two-electron bonding due to electron deficiency. That is why the energetically favorable structure of borophene is a mixed hexagonal (electron deficiency)-triangular (electron surplus) lattice (e.g. $V_{1/6}$ and $V_{1/5}$ phase), which leads to inequivalent coordination numbers, such as 4, 5, and 6 in the $V_{1/6}$ phase. According to the TERS and theoretical studies shown in Figure 2 and 4, bridge sites in either $V_{1/6}$ or $V_{1/5}$ structures, which link two boron atoms with a coordination number of 4, are the most vulnerable to oxygen attack. This implies that π bonds formed between the boron atoms with low coordination numbers are susceptible to chemical modifications, such as H, F, Cl functionalization (Phys. Chem. Chem. Phys. 2019, 21, 7630; Comput. Mater. Sci. 2019, 156, 56) or to interlayer bonding. Consequently, it suggests that chemical passivation or the formation of multi-layer structures via those active sites could enhance the chemical stability of borophene. We appreciate the reviewer's insight into this issue and admire his/her foresight, as we're happy to see that the improvement of ambient stability of borophene by the above-mentioned approaches has been experimentally demonstrated in the recent past

months. Li *et al.* reported the hydrogenation of borophene via atomic hydrogen in UHV, where the H atoms chemically adsorb to the B–B bridge sites with three-center–two-electron B–H–B bonds (Science 371, 1143–1148). The resulting hydrogenated structure (i.e., borophane) showed enhanced chemical stability (negligible oxidation) even after 1 week of ambient exposure. Another work reported on the synthesis of bilayer borophene which features covalent interlayer B–B bonding via those boron atoms on the ends of bridge sites. The bilayer borophene is also inert to ambient oxidation compared to the monolayer counterparts, resulting from the significant charge transfer and redistribution due to the interlayer covalent bonding.

Revision: We have added a brief comment in the Conclusion section to address the reviewer’s concerns.

“We noticed that boron atoms with low coordination numbers (e.g., 4 in $v_{1/6}$ or $v_{1/5}$ phase) and the bridge sites linking them show remarkable activity to atomic adsorption (e.g., H, O, F) via multi-centered-two-electron bonding or to the formation of covalent interlayer bonding (in multilayered borophene). This insight has been demonstrated by the recent reports on the synthesis of hydrogenated borophene³⁹ and bilayer borophene⁴⁷ and their enhanced inertness to ambient oxidation due to the chemical passivation of active sites via B–H–B bonding or charge transfer and redistribution through covalent interlayer B–B bonds.”

(2) Is there a dose limit for recovery after oxidation via atomic O or molecular O₂ exposure? Or restated differently, how much oxidation is fully destructive to the borophenes? The oxidation of borophene, and ostensibly the modification of the intrinsic properties, are a strong barrier to further applications and straightforward device fabrication. It would be great to explore whether the borophene can be recovered from intense oxidation via a similar protocol detailed in Fig. 6.

Reply: There is no dose limit for recovery after oxidation via atomic oxygen due to the predominance of chemically uniform surface species (O adatoms). In contrast, even a low dose of O₂ (e.g., typically 1200 L) can lead to heterogeneous oxidized surface decorated with various boron oxide clusters, which cannot be thermally reduced and remain on the surface even after annealing up to 400 °C. The protocol described in Fig. 6 can reduce some complex boron oxides to simple species (e.g., oxygen adatoms), providing an opportunity to recover the pristine borophene surface for practical applications. Unfortunately, it is a daunting task to apply this approach to a variety of surface species on an intensely oxidized and chemically inhomogeneous sample, as (1) the manipulation conditions are highly dependent on the chemical structure and adsorption configuration of surface species; (2) tip-induced reactions take place very locally and are hardly performed on a large scale that is appropriate for device fabrication; (3) the case will be more complicated when the sample is exposed to ambient conditions due to the presence of water.

Reviewer #3 (Remarks to the Author):

In this manuscript, the authors used ultrahigh vacuum tip-enhanced Raman spectroscopy (UHV-TERS) to chemically interrogate individual oxygen adatoms on a boron monolayer, which is helpful for the deep understanding of the atomic-scale reactions. It is nice to see that the authors successfully manage to manipulate single atom and induce the single atom reaction with the substrate, and to chemically identify the formation of the bond via vibrational fingerprint. The article can be published after addressing the following questions:

Reply: We appreciate the reviewer's very positive comments and recommendation of publication. We respond to the questions below and have revised the manuscript accordingly.

1. While discussing Figure 2, the authors described Figure a first, followed by Figure d and then Figure b, c (d comes before b and c, which is awkward). Thereafter, Figures e-g are described in the next paragraph. Such a logic is awkward. I highly suggest the authors to describe the phenomenon in Figure a first thoroughly, followed by simulation of the most stable structure and the possible vibrational modes. Then, the authors can try to correlate their result of simulation results.

Reply: We appreciate the reviewer's suggestion very much and would like to rearrange the figures and revise the paragraph accordingly.

Revision: We have rearranged Figures 2b-g and changed the sequence of the sentences describing Figures 2a-g, which now read more logically (yellow highlighting on Page 4 in the revised manuscript).

2. In Figure 2i, from the description, it seems the authors assign both 205 and 189 cm^{-1} modes to the adsorbed single oxygen adatom. If this is the case, their trends of the TERS intensity profiles should not be inverse. I assume the authors would rather mean the 189 cm^{-1} mode comes from the mode of borophene under the influence of the neighboring O atom. If this is the case, the authors should clearly state this in the main text. Furthermore, the vibrational modes shown in Figure b-d can not be easily read. I would suggest the authors to increase the size or enlarge a portion of the model.

Reply: We agree with the reviewer that if the 189 cm^{-1} mode solely originates from the adsorbed atomic oxygen (i.e., B-O bonding), it should show the same (instead of inverse) trend of intensity profile as that of the 205 cm^{-1} mode. However, as the reviewer speculated, the observed 189 cm^{-1} mode actually includes the mode of borophene lattice under the influence of the nearby O atom as demonstrated in Figure 2a, which shows a low intensity at the O atom site due to the higher tip position therein. In order to further clarify this issue, we would like to add a sentence in the main text.

In addition, following the reviewer's suggestion, we would like to increase the size of schematic models of Raman modes shown in Figure 2.

Revision: We have added a sentence "Note that the evolution of the 189 cm^{-1} mode reflects the Raman intensity change of both bare and O-adsorbed $\nu_{1/6}$ borophene" in the paragraph starting with "To see exactly how TERS spectra evolve across an oxygen adatom".

We have also increased the size of atomic models of Raman modes in the rearranged Figure 2.

3. In the 2D TERS intensity mapping shown in Figure 2k and i, the authors mentioned that the oval-shaped rather a circle came from the thermal drift during TERS collection. To validate this assumption, the authors should provide the drift value of this TERS system, so that the result in the image distortion can be estimated.

Reply: Typically, our TERS system has a thermal drift of 0.15–0.2 nm/min in X and Y directions under laser. However, the drift value is highly dependent on laser power (4–10 mW for the present research system), so sometimes it could be beyond the above-mentioned drift range.

Revision: We have added the typical drift value in the manuscript.

4. In Figure 3d, the cluster shows no 205 cm^{-1} features in the 2D TERS mapping. It is better to also show the complete TERS spectra on the cluster site and the oxygen adatom site.

Reply: We appreciate the suggestion and add the data into Supplementary Information.

Revision: We have added the TERS spectra acquired on the cluster and oxygen adatom sites in the resubmitted Supplementary Information (Supplementary Figure 8).

5. When comparing the activity of borophenes of different phases, why different amount of oxygen was introduced into the system (3600 L and 4800 L)?

Reply: We thank the reviewer for pointing out this detail. On the one hand, we did not attempt to quantitatively compare the reactivity of $\nu_{1/6}$ and $\nu_{1/5}$ phases to oxygen. Instead, we just wanted to highlight the qualitative differences in the oxidative behavior of the two phases (uniform vs selective oxygen adsorption). On the other hand, higher O_2 dose leads to more adsorbed oxygen along the line defects within $\nu_{1/5}$ domains, thus giving rise to sharper contrast to the uniformly distributed oxygen on the $\nu_{1/6}$ phase.

6. Were Figures 5c and d obtained from the same position? I guess they are from different positions, as the substrate has undergone annealing. If this is the case, the authors should clearly state they are from different position, otherwise, the comparison of Figures 5a and b will mislead the understanding of c and d.

Reply: We appreciate the reviewer's attention to detail. Figures 5c and d were acquired from different areas. We have revised the caption of Figure 5 to address this issue.

Before revision:

"c,d, STM images of O-oxidized borophene before (c) and after (d) annealing at 360 °C."

After revision:

"c,d, STM images of O-oxidized borophene before (c) and after (d) annealing at 360 °C, which were acquired from different surface areas."

7. In figure 3b, why the two borophene with $v_{1/6}$ phase showed two different contrast?

Reply: Figure 3b shows a borophene island continuously growing over a Ag(111) step edge. Therefore, the borophene sheets grown on upper and lower Ag terraces show different height contrast. We have added a phrase in the caption of Figure 3b to clarify this issue.

Before revision:

"b, STM image of a $v_{1/6}$ borophene island after exposure to 3600 L of molecular oxygen."

After revision:

"b, STM image of a $v_{1/6}$ borophene island across a Ag(111) step edge after exposure to 3600 L of molecular oxygen."

8. It is not stated but can be observed that oxygen clustering tends to form near borophene edges, which need to be stated and explained.

Reply: We thank the reviewer for reminding us to highlight this effect. In the paragraph starting with "UHV oxidation of borophene with molecular oxygen.", we described this phenomenon by stating that "In contrast to the topographically and chemically sharp edges of pristine borophene, high O₂ doses led to a significant morphological degradation of borophene edges by forming inhomogeneous oxide species, indicating the high sensitivity of borophene edges to molecular oxygen (see additional STM and TERS characterizations in Supplementary Fig. 6)."

In addition, we addressed this issue in the first paragraph of the Supplementary Information where we stated that "This results from the enhanced lateral mobility of atomic oxygen at elevated temperatures and the strong thermodynamic and kinetic driving forces that lead to oxygen adsorption and boron oxidation at borophene edges due to rich dangling bonds therein."

9. *The authors state that the simulations were performed based on a freestanding borophene monolayer without taking into account interactions with the Ag substrate, which led to the mismatch between theoretical values and experimental measurements. So, I am wondering are there any challenge in phonon simulations to include the effect of the Ag substrate.*

Reply: If including the Ag substrate, the computational system will become too large (with too many atoms), which is beyond our current computational capability.

10. *In Figure 6c, when the bias is changed to 4.0 V, why the corresponding oxidized borophene height shows an immediate change at one line rather than changes from the beginning position?*

Reply: Boron oxide clusters formed on oxidized borophene could decompose under the bias of 4.0 V or higher. This tip-induced decomposition reaction is dependent on the chemical structure and adsorption configuration of oxide clusters. However, the oxide species formed on O₂-oxidized borophene are chemically inhomogeneous, as demonstrated in Figure 3. Therefore, when and where the decomposition reactions begin are highly dependent. For example, Supplementary Figure 13 (in the resubmitted Supplementary Information) shows a second decomposed oxide cluster and the reaction occurs near the beginning position (the edge of the cluster). In addition, we found that the third cluster failed to react under the same tunneling conditions.

11. *The three isolated oxygen adatoms always exist in Figure 6a, e and f. It seems that the corresponding STM images show no features of these three adatoms when the bias is changed. Then, they will appear at the same position when the bias returned to 1.3 V. The authors should provide more explanations and the contrast bar of these STM images which is helpful for reading.*

Reply: We thank the reviewer for the valuable suggestion. As revealed in Figures 1e-g and particularly in Supplementary Figure 1, the topography of oxygen adatoms on borophene is highly dependent on sample biases. Specifically, STM images show no features for oxygen adatoms when the scanning bias is 3.5 V or higher (Figure 1e and Supplementary Figure 1), while they show oxygen adatoms as protrusions under 2.2 V (Figure 1g and Supplementary Figure 1). That is why the oxygen adatoms seem to appear and disappear when the sample bias is switched, which is consistent with our demonstration in Figure 1.

In order to avoid confusion, we have highlighted that “(Fig. 6b; oxygen adatoms are invisible due to the bias dependence of STM morphology)” in the paragraph starting with “Local interrogation of the stability of oxidized borophene”.

Furthermore, we would like to add a sentence in the caption of Figure 6 to address this issue and thus facilitate the understanding of Figure 6.

Revision: We have added a sentence in the caption of Figure 6: “The absence of oxygen adatoms in (b-d) is due to the bias-dependence of STM topography rather than tip-induced desorption”.

12. In Figure 6g, the author showed the scheme that there will be some clusters remaining on tip after the STM manipulation. We suggest the authors to check the contamination on the Ag(111) surface instead of clean borophene, which will be more convincing. The authors should also show the spectra of after tip retraction in all TERS results, which can convincingly demonstrate that the collected Raman signals are indeed from the sample instead of clusters on the tip.

Reply: We are grateful to the reviewer for his/her important suggestion about the details of TERS measurements. Actually we do check the cleanness of TERS tip on clean Ag(111) surfaces before and after each TERS measurements, which we also think is more convincing. We would like to present the corresponding TERS spectrum acquired from the nearby Ag(111) surface in Supplementary Information. In addition, following the reviewer’s suggestion, we would like to add the spectra collected after tip retraction for all TERS results throughout the manuscript.

Revision: We have presented the spectrum acquired on Ag(111) and the spectrum collected when the tip was retracted in the resubmitted Supplementary Information (Figure S14), and added a sentence in the manuscript “Additional evidence is the featureless Raman spectrum acquired on the nearby Ag(111) surface under the same condition and the absence of Raman signals when the tip was withdrawn (Supplementary Fig. 14).”

Moreover, we have added the spectra collected after tip retraction for all TERS spectra throughout the manuscript.

REVIEWERS' COMMENTS

Reviewer #2 (Remarks to the Author):

The authors have addressed my comments. I recommend publication of this noteworthy manuscript.

Reviewer #3 (Remarks to the Author):

The authors have satisfactorily addressed my points. I would suggest to accept it.